# Auto-Cucumber: The Impact of Autocorrection Failures on Users' Frustration

Ohoud Alharbi*
King Saud University

Wolfgang Stuerzlinger†
Simon Fraser University

## ABSTRACT

Many mobile users rely on autocorrection mechanisms during text entry on their smartphone. Previous studies investigated the effects of autocorrection mechanisms on typing speed and accuracy but did not explore the level of frustration and perceived mental workload often associated with autocorrection. Through a mixed-methods user study, we investigate the effect of autocorrection failures on increasing the user's frustration, mental and physical demand, performance, and effort in this paper. We identified that perceived mental and physical demand, and frustration are directly affected by autocorrection.

**Index Terms:** Human-centered computing—Interaction design and evaluation methods—Keyboards;

## 1 INTRODUCTION

Empowered by the growth of text-based social media, many people prefer writing text messages or social media posts over making phone calls. To keep up with this growth, text entry methods have been improved by providing features that enable users to type as fast as possible and correcting their typing errors as they go. Yet, being fast and accurate can be a challenge on touch screen keyboards, due to various issues, including misspelling the word, using the wrong touch locations, missing a space, and compounded versions of these.

Still, a frustrating interaction with a computing device, resulting from typing errors or a wrong autocorrect, can cause users to experience negative emotions toward the system and to potentially abandon using some functionality [30]. In that moment of frustration, users might not be aware how much autocorrect has already improved and keeps improving with continuous use and upgrades to algorithms. To better understand the origins of current user reactions, this paper focuses on an analysis of the behaviors people exhibit in text entry with respect to autocorrect and its failures and the associated costs in terms of perceived mental and physical demand, and user frustration.

Text entry research typically collects data to evaluate the speed and accuracy of a new interaction technique, such as Drag-n-Drop, Drag-n-Throw, and Magic Key [53]. Studies have examined the effect of keyboard layouts on typing behavior, e.g., [6,19,21,28,49], while other studies have investigated the time users spent while interacting with autocorrections and the prediction panel while entering text, including when prediction and autocorrect approaches fail, e.g., [1,2,10]. However, there are no studies that investigate the effect of failing autocorrections on the user's emotions and their level of frustration. Yet, cognitive theory research has shown that system failures can activate negative emotions such as anger, annoyance, and frustration [35].

This paper presents a user study that investigates the effect of various degrees of failing autocorrection on the user's frustration and perceived mental workload. We analyze the results through metrics

---
*e-mail: omalharbi@ksu.edu.sa
†e-mail: w.s@sfu.ca

related to individual keystrokes, but also use qualitative methods, such as survey questions, observations, and interviews. After a discussion of related work, we present the results of our study (N = 20) to observe the effect of failing autocorrection on users' mental workload. Results show that perceived mental and physical demand, and frustration levels are affected by autocorrection. There is a need to further investigate ways to give users the ability to temporarily adjust the behavior of autocorrection without turning this generally beneficial feature permanently off. Based on user feedback, we propose mechanisms such as adding a (single-step) button on the keyboard to quickly toggle autocorrection, or displaying a confidence score at the side of the screen.

## 2 RELATED WORK

Frustration can lead users to believe that they are failing a task [7]. Further, a frustrating interaction with a computing device can cause users to feel negatively toward the system and then encourage them to potentially turn off some aspects of its functionality, such as autocorrect [30]. If feelings of frustration are strong, they may even make a user abort or re-consider an action [46]. For instance, excessive download delays might have a negative impact on the brand perceived to be responsible for the delay [42]. Feelings of frustration are linked to the perceived duration of activities [8, 17]. There is much potential negative impact when users are frustrated and unable to respond to failures or give feedback [35].

Nevertheless, it is not always the case that negative emotions will increase as failures occur more frequently. While there will generally be a negative emotional response to failure, there may also be a lowering of expectations, which will tend to make emotional responses to subsequent failures less intense [36, 44].

### 2.1 Predictive Features

As errors contribute substantially to slow real-life text entry speed, facilitating error correction is a key challenge for text entry [26]. Errors are costly in time and effort, and can negatively affect user perception of text entry quality. Yet, the visibility of errors and suggestions for error correction can also increase both perception and interaction costs, which might even reduce text entry speed, e.g., [27,32,39,40], and in some cases decrease writing accuracy [4]. Previous work has identified that word correction and completion features on mobile keyboards could save up to 45% of keystrokes [16], but this promise rarely results in a corresponding increase in typing speed [15].

If an appropriate language model is used, predictive algorithms can support effective error correction and completion [16]. However, many other factors play a role in the effectiveness of the use of predictive features [31], including the experience of the user [38]. To enable us to study the effect of failures in a systematic manner and how users experience such failures, we strategically caused the autocorrection to fail with controlled frequencies in our study.

### 2.2 Frustration and Mental Workload Assessment

Workload is a term used to characterize the effort associated with a job and refers to the amount of work that needs to be performed ('the work'), usually within a fixed period of time ('the load'). Mental workload is the level of measurable mental effort put forth by an

individual in response to one or more cognitive tasks [52]. We can assess mental workload using physiological or self-report measures.

Physiological measures used to measure mental workload can include frustration, since these feelings are accompanied by physiological changes. Ceaparu et al. [8] measured the physiological response associated with workload by simulating frustrating experiences that someone might have when they play a game. At specific intervals the mouse would fail, leading to frustration. Yet, emotional experiences may be influenced by many factors such as individuals' memory, life history, culture, age, and gender [25]. More research is thus needed to identify how different physiological methods, e.g., skin conductance and heart rate variability, can be combined to develop more objective measures of frustration that are both effective and reliable. Still, we believe that physiological measures are currently not yet reliable enough to be used as a main measure of frustration.

Alternatively, self-reports are a subjective assessment that rate perceived workload to assess a task, system, or other aspects of performance. With this approach, researchers ask participants to rate their response after an intervention or interruption.

To compare self-reports with Physiological measures, Cooper et al. [9] evaluated four sensors in terms of utility for frustration research: a camera that focused on the participants' face, a skin conductance bracelet, a pressure sensitive mouse, and a chair seat capable of detecting posture. Participants were presented with questions such as "how [interested/excited/confident/frustrated] do you feel right now?" and rated their current state on a scale of 1 to 5. The authors found that the most accurate results came from the self-reported assessment.

Further, the NASA TLX is a popular and well-validated self-report questionnaire to measure the experienced workload and was initially developed to measure workload in the military [23]. It has been applied in a variety of settings in human-computer interaction research [11]. The NASA TLX combines six scales, including mental demand, physical demand, effort, and frustration.

Frustration is an important component of mental workload. Many researchers developed questionnaires to specifically measure this emotion. Ceaparu et al. [8] forced a frustrating situation and asked participants to subjectively report on each frustrating experience, once it occurred during the session. Van Steenburg et al. [47] and Gelbrich [18] developed questionnaires that measure frustration in an imagined frustrating situation. Goldsmith et al. [20] developed an online questionnaire including scales that measure attitude and frustration tolerance [22]. Richins [41] used a method based on ratings of seven frustration-related adjectives (frustrated, uncomfortable, anxious, stressed, strained, annoyed, and awkward). Similarly, Wu and Lo [50] developed ten items aimed at measuring how a telecommunications service is performing relative to customer expectations. Droit-Volet and Wearden [14] measured the mood of participants throughout the day using an experience-sampling method or short survey.

The approach of repeatedly measuring mood states has been used in a number of further studies [12–14]. Since repeatedly using self-report measures is a standard method in human-computer interaction research, we decided to adopt this approach by repeatedly measuring workload and frustration states through a short survey based on the NASA TLX questions.

Finally, the complementary combination of self-reported measures together with qualitative analysis can yield an even better representation of a users' mental state [24]. For instance, an exploratory study [24] employed questionnaires, think-aloud protocols, and in-depth interviews to determine the primary points of critique and satisfaction with the information provided on a website, by examining the properties of the website, the search process, and the mood alterations of the participants in combination. Using the think-aloud method often provides good explanations about the users'

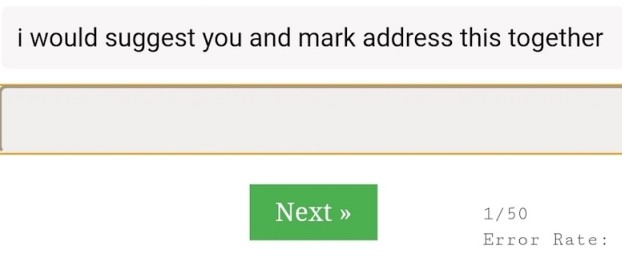

Figure 1: The webpage that participants saw during the experiment.

thought process and reveals changes of mood [37].

### 2.2.1 Motivation and Experimental Approach

Our work aims to highlight the potential side-effects of "smart" techniques that are automatically applied, such as autocorrection. We investigate the effect of failing autocorrections on the user's level of frustration and perceived mental workload. Based on the above review of methods to measure frustration and mental workload, we decided to combine different methods to arrive at a more complete picture of the outcome. Following previous work [24], we combine self-report questions with qualitative protocols, more specifically think-aloud and interviews, to better understand the reactions of our participants. According to previous work, this approach currently still yields a better representation of users mental states than using physiological measures [24]. We also follow Ceaparu et al.'s [8] approach by forcing a frustrating event (autocorrection failures) and asked participants to subjectively report on their experience during the session.

### 3 APPARATUS

We used a web application for data collection. We implemented the system using HTML, CSS, JavaScript, and PHP. We then used Amazon Web Services (AWS) to host our web application. The application includes a custom autocorrection method that works independent of various operating system implementations. The system presents prompts with text for the user to enter and logs all occurring events at the keystroke level (Figure 1).

### 3.1 Instructions

Participants initially needed to acknowledge that they had read the instructions and to also give their consent for data collection. These initial instructions asked participants to temporarily disable the predictive features on their phones. Once participants agreed to participate, they were instructed on the procedure and then started the English language text entry tasks. The main part of the experiment showed only a single line of instruction, a presented phrase, and a textbox to input that phrase, see Figure 1, as well as the user's own keyboard, which they used for text entry. We asked participants to use their own device and their own keyboard layout, because we wanted to eliminate the associated learning factor and any potential influence of such learning on their frustration. Users needed to tap on the "Next" button to move to the next phrase, where they then also saw an up-to-date average for their text entry speed and

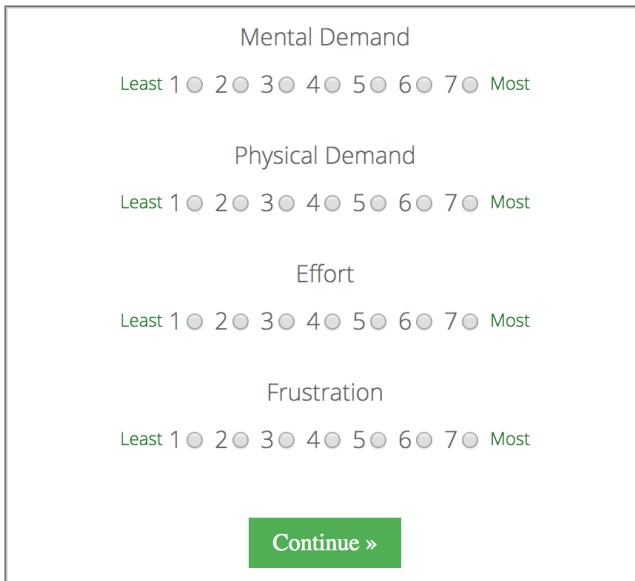

Figure 2: Our short survey to probe frustration, effort, and mental and physical demand.

error rate. In between blocks of 5 phrases, participants were presented with questions about how much mental demand/physical demand/effort/frustration they felt at that moment, rated on a scale of 1 to 7, see Figure 2. We purposely removed the questions regarding temporal demand and performance from the NASA TLX, since in our instructions we asked participants to type as fast as possible and to maintain a low error rate. These questions appeared before the task and were then shown each time after the users had entered 5 phrases. Following previous work, we asked the users to answer the questions repeatedly to better understand the contingencies of their behavior [12–14]. We used transcription typing to measure participants' typing speed, as this approach enables us to study motor performance while excluding cognitive aspects related to the process of text generation [38].

### 3.2 Custom Autocorrection

To ensure that we could correctly log every text entry action, we asked participants to disable their own predictive system, including their prediction panel and autocorrection. Another reason for this decision was that we needed to manipulate some internals of the autocorrection mechanisms in our study, something that current system APIs do not permit. We thus used a custom autocorrection algorithm that gets triggered when an inputted word does not match the word in the presented text.

For autocorrection we exposed participants to four different conditions: optimal, failures 10% of the time, failures 20% of the time, and no autocorrection. In the optimal condition, if the misspelled word is close enough to the intended one, our system autocorrects it to match the presented word. This conditions always produces perfect autocorrections, which is similar to the "100% accurate" autocorrect condition in [5]. This condition closely resembles an oracle.

For autocorrection that fails 10% (20%) of the time, we adjust the system to produce a correct autocorrection 90% (80%) of the time (using the optimal method), but produce only a "close-enough" result in the remaining 10% (20%) of the time. To create such an almost correct result, our implementation searches for similar words using the Levenshtein distance [33] and then chooses the one with the lowest editing distance, i.e., a word that looks like a plausible autocorrect. We used a dictionary with the 40,000 most frequent words from project Gutenberg[1]. We verified that our prediction algorithm matches commercial systems reasonably well. For this, we randomly chose phrases and compared the output of our system with that of an Android 9 keyboard using the same input test. We found that the output matches 94% of the time, which is reasonably high and likely at a level that is not easily perceived to be different by naive users.

### 3.3 Data Logging

Through our web-based system, we recorded each text change or touch event, which fairly closely corresponds to the keystroke logging level, with a corresponding timestamp. For each phrase, we recorded the following data: device orientation (portrait/landscape), presented text, typed text, the complete input stream, keystrokes per character, words per minute, and total time per phrase. Moreover, we also logged all autocorrections, cursor movements, and error messages that were triggered during text entry. This comprehensive logging enables us to fully replay the input of each phrase.

### 3.4 Phrase Set

We used 30 phrases randomly selected from the Enron MobileEmail phrase set [48]. We removed all non-alphabetic characters, including punctuation, and made sure that the selected phrases contained at least three words. We decided to exclude non-alphabetic characters and punctuation in the study, as such characters introduce a potential confounding source of variation in the dependent measures and threaten internal validity [34]. The phrases in the set (774 sentences) were generally short to medium length, average 6.1 words (SD 1.68, ranging from 3 to 12), and contained on average 29.9 characters (SD 10.13, ranging from 14 to 67).

## 4 USER STUDY

The purpose of this study was to compare 4 conditions of autocorrection (optimal, failing 10%, failing 20%, and none) and to measure the associated perceived mental and physical workload of the user. Previous work identified that the largest error rate at which typists would attempt to type before autocorrect corrects errors ranges between approximately 15% and 25% [5]. In our pilot studies, we initially experimented with conditions that exaggerated the number of failures (up to 40% failures on autocorrects). Yet, we observed that high error-rate conditions (larger than 25%) were extremely confusing for participants. Thus, we decided to exclude such conditions from our main study and to examine only the 10% and 20% options. With similar conditions, we also ran a pilot study with a within-subject design and found indications for a substantial carryover effect that influenced participants' answers, based on the sequence in which the conditions appeared.

### 4.1 Design

We used a between-subjects design. Each participant entered 30 phrases with one of the 4 conditions (no, 20% failing, 10% failing, and optimal autocorrection), excluding two practice phrases. In total we collected (20 participants × 30 phrases) = 600 phrases.

### 4.2 Procedure

Before starting this study, participants were asked to complete a background questionnaire about their age, gender, English proficiency, and their experience with their current touchscreen device keyboard, including what they thought about the performance of their current autocorrection system. We also gave them a full demonstration of our system and let them experience text entry using it for entering a few training phrases (using the chosen condition for

---

[1]https://en.wiktionary.org/wiki/Wiktionary:Frequency_lists

that participant, i.e., if their assigned condition was optimal auto-correction, they experienced this already in the training). During the study, participants were asked to enter 30 English phrases using our system and to answer questions about how much mental and physical demand, effort, and frustration they felt at the moment, see Figure 2. Each participant answered the questions seven times, once before the typing task started and the remaining six times after entering each block of five phrases. Additionally, we asked them to use the think-aloud method, which we explained to them during the training phase.

At the end of the session, we conducted a semi-structured interview targeting behaviors we had observed or comments users had made during the text entry sessions. Further, we also asked participants about their own stories around autocorrection, i.e., positive or negative episodes that they had encountered in the past. We also asked them about how they believed that autocorrection influenced their typing speed and correctness, and how autocorrection made them feel. Other questions inquired about the type of words that they find hardest to get correct with current autocorrect systems and finally if they had any design recommendations around autocorrection.

Including signing consent forms, filling questionnaires, the main typing tasks, and the interview, the session lasted about 45 minutes on average. We used two cameras and tripods, as well as voice recording to assist observation. Figure 3 shows the setting of the experiment. One camera was directed at the mobile screen and the second at the participants' face to record their expressions. The user study was approved by the research ethics board of the local university.

### 4.3 Participants

We recruited twenty participants (10 females, 10 males) for the study through advertising to a student participant pool at Simon Fraser University. Of these participants, 14 were between 18 and 24 years old and 6 between 25 and 34. Half of the participants indicated that they are using a mobile keyboard with Latin characters, i.e., the modern English alphabet, constantly during the day, 30% more than once per hour, and 20% more than once a day.

Even though our task did not require high English proficiency, we created a quick English quiz using material from http://iteslj.org towards an objective assessment of English skills. The "overall success rate" was the final score participants achieved in our language proficiency quiz that consisted of six grammar questions: two easy, two medium, and two hard. Results show that the success rate for the overall English proficiency quiz was 92% (SD = 13), which corresponds to reasonably high English proficiency, as is to be expected for a university environment. Given this level of proficiency, we did not follow up on this data.

Among our participants 65% used Android or variants (Samsung, OxygenOS, etc.), while 35% used Apple iOS. Most (90%) indicated that they normally have autocorrection activated on their devices. When we asked them to rate predictive features in their mobile devices on a 5-point Likert scale (very good, good, acceptable, poor, and very poor), 5% chose very good, 55% indicated good, 35% acceptable, and 5% very poor.

### 5  RESULTS

We used one-way ANOVA with alpha of 0.05 for all analyses. A Shapiro–Wilk test identified that the assumption of a normal distribution was satisfied, and all other preconditions of ANOVA were also met. We used Tukey's Honest Significant Difference (HSD) test for post-hoc analyses. To characterize effect sizes we used the partial eta squared measure.

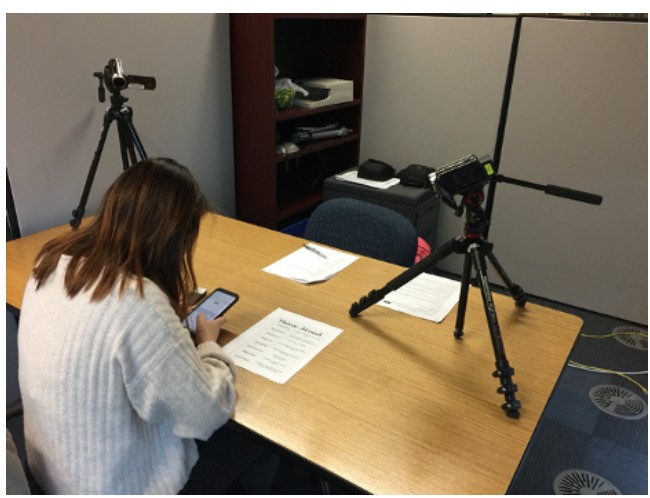

Figure 3: The experiment setting.

#### 5.0.1  Performance

In line with common text entry study protocols, we used the words per minute (WPM) metric to measure entry speed [3, 45]. Time was measured from the first keystroke to the last. **We observed a statistically significant effect on entry speeds for the four conditions**, $F(3,136) = 3.491$, $p < .018$, with optimal being the fastest option, with a medium effect size $\eta_p^2 = .07$ see Figure 4.

We also measured the verification time, i.e., the "reviewing time", which is the time participants took to review a phrase before moving to the next. For this, we measured the time from the last keystroke until the time participants pressed the "next" button. **Verification times were statistically significantly**, $F(3, 136) = 3.51$, $p = .04$, with a large effect size $\eta_p^2 = .4$. Optimal and 10% autocorrection required less verification time, see Figure 4.

The difference in terms of the number of **keystrokes per character (KSPC) for each condition was statistically significant** [3, 45], $F(3, 136) = 4.97$, $p = .013$, with a large effect size $\eta_p^2 = .48$. No autocorrection had higher KSPC as shown in Figure 4, corresponding to more keystrokes spent on error correction.

We analyzed the average Error Rate (ER) of the final submitted text, and found it was not significantly different across conditions, $F(3, 136) = 2.256$, $p = .085$.

We further investigated the use of error correction methods, such as the number of backspaces and cursor movements. We found the use of backspaces to be statistically significant, $F(3, 136) = 5.39$, $p = .009$, with a large effect size $\eta_p^2 = .5$, but the use of cursor movements is not significant $F(3, 136) = 2.36$, $p = .11$. Optimal and 10% autocorrection required fewer backspaces.

The average rate of autocorrections events that occurred due to participants making typing errors with the 20% failing condition were M = 12.60% (SD = 18.29), for 10% failing M = 10.51% (SD = 5.47), and for the optimal condition were M = 17.99% (SD = 14.70). Of those recorded events, the average percentage of forced failures, i.e., where the system simulated a failure, were 19.66% (SD = 14.44) for the 20% failing condition, 7.5% (SD = 5.01) for the 10% failing condition, and 0% for optimal condition.

#### 5.0.2  The NASA Task Load Index

**We observed a statistically significant effect on frustration**, as measured by the corresponding question from the NASA TLX, $F(3, 136) = 12.686$, $p < .001$, with a large effect size $\eta_p^2 = .22$. Optimal stood out by being the least frustrating. There was also **a statistically significant effect on mental demand across conditions**,

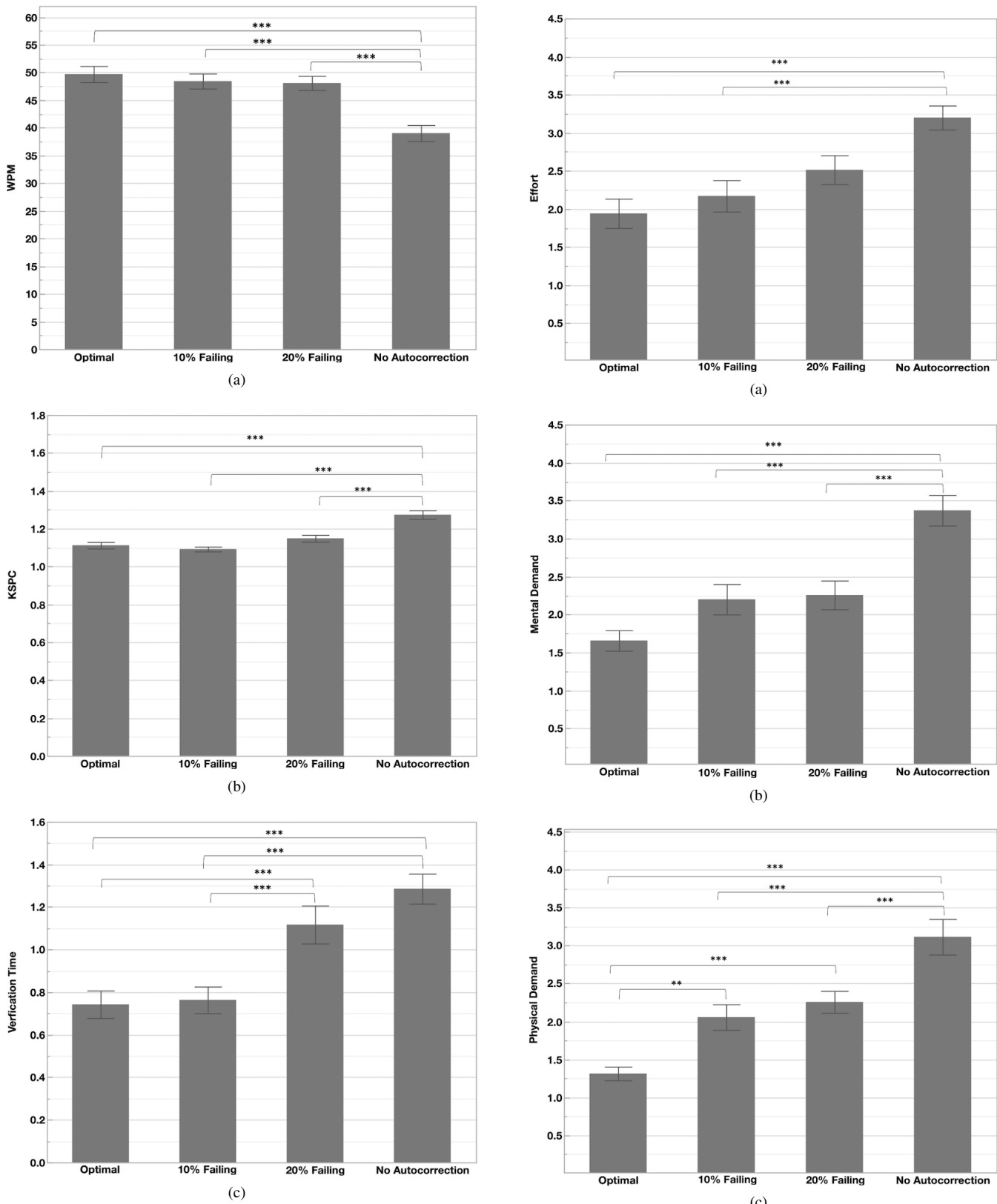

Figure 4: a) Average words per minute (WPM), b) average keystrokes per character (KSPC), and c) average verification time for each condition (seconds). The three asterisks (***) illustrate a significant difference with $p \leq 0.001$.

Figure 5: a) Average effort, b) average mental demand, and c) average physical demand for each condition. The three asterisks (***) illustrate a significant difference with $p \leq 0.001$.

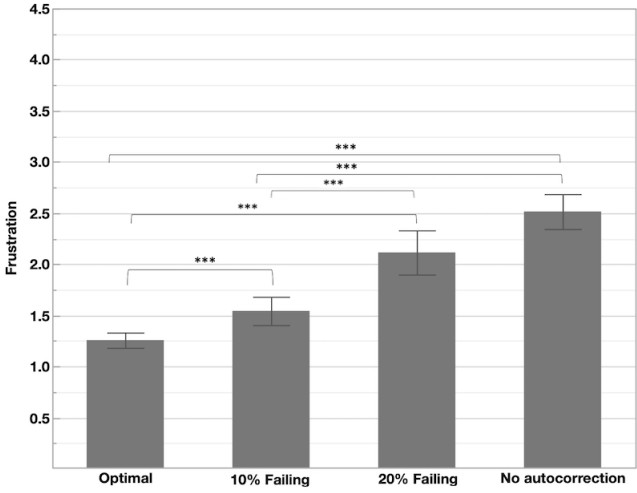

Figure 6: Average frustration for each condition. The three asterisks (***) illustrate a significant difference with $p \leq 0.001$.

$F(3, 136) = 15.361$, $p <.001$, with a large effect size $\eta_p^2 = .25$. No autocorrection was significantly more mentally demanding. Additionally, we observed **a statistically significant effect on physical demand across conditions**, $F(3, 136) = 19.51$, $p <.001$, with a large effect size $\eta_p^2 = .30$. Here, no autocorrection was followed by 20% autocorrection as being the two most physically demanding conditions. Finally, we observed **a statistically significant effect on effort across the conditions**, $F(3, 136) = 8.55$, $p <.001$, with a large effect size $\eta_p^2 = .16$. No autocorrection and 20% autocorrection required most effort. The means and results from the post-hoc analyses are presented in Figure 5. As we had prompted participants with our survey seven times during the study to investigate changes in frustration and workload over time, we illustrate the fluctuations of the answers in Fig. 7.

### 5.0.3 Interviews

At the end of the session, we conducted a semi-structured interview with each participant, focusing on any observed behaviors or comments users made during text entry. We analyzed what people told us, by first coding our interview data in a systematic manner and then identifying larger themes from that data.

When we asked participants about their experience with autocorrection, Participant 4 mentioned that it offers easy help to accelerate typing. Participant 6 stated, *"It helps me type so much faster than all my friends because they don't use it. So, I would say almost all of the time, [it] is a good experience,"* and Participant 2 said, *"it's a mini helper."* Still, Participant 11 indicated that it can slow them down, disturb, and hinder the communication. Participant 13 had a more balanced view and said, *"It can be helpful, but also detrimental."*

We also asked for stories about (positive or negative) episodes that participants had encountered with autocorrection. Participant 5 said, *"my friend was complaining about autocorrect in a text and it was changed to 'auto cucumber'.",* which was humorous enough to make it into the title of this paper. However, Participant 7 said, *"a friend of mine sent an entirely different text to his wife because of autocorrection. She was so mad. He had to [provide] a lot of explanation to calm [her] down."* Autocorrection also can lead to social embarrassment, as Participant 11 said, *"due to autocorrection [I] typed a slang [word] instead of a person's surname. This was on a WhatsApp group chat. Later people mention this personally and I was so embarrassed,"* while Participant 13 said, *"while messaging in a family group, autocorrection changed my wishes from*

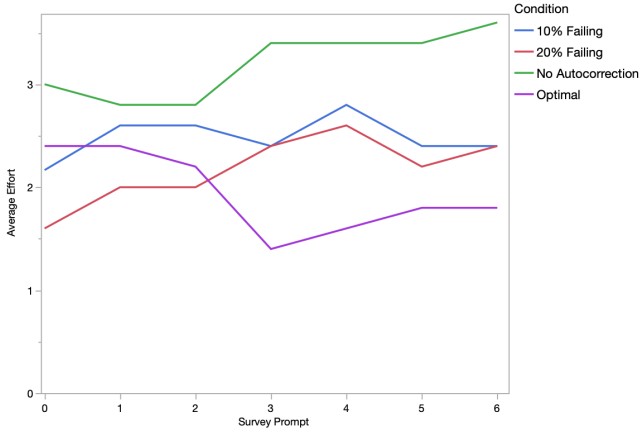

(a)

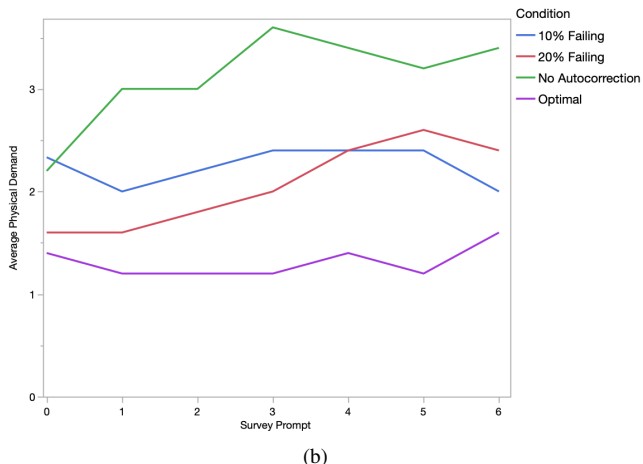

(b)

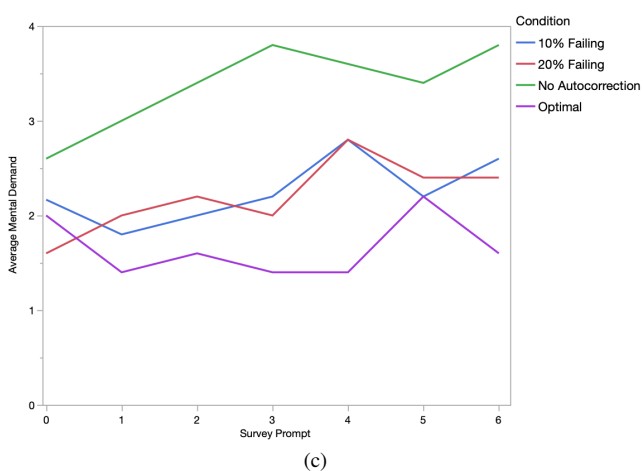

(c)

Figure 7: Average a) effort, b) physical demand, and c) mental demand for each condition for each survey prompt starting from the initial baseline prompt.

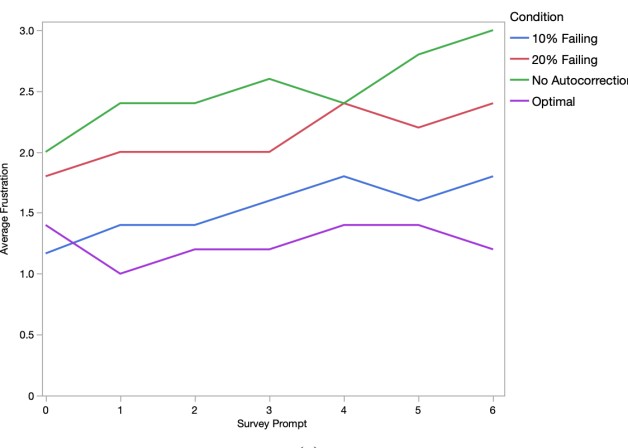

(a)

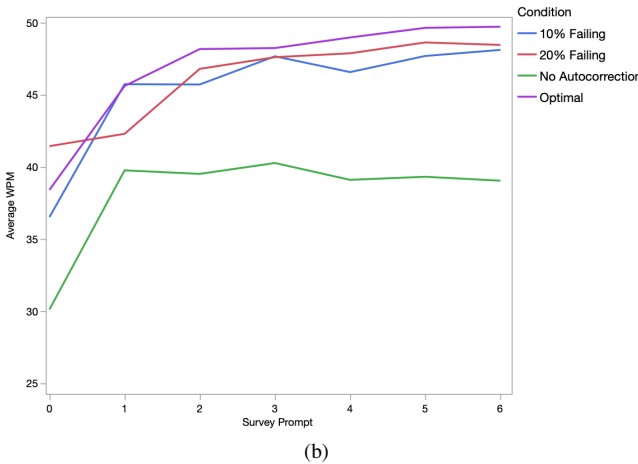

(b)

Figure 8: Average a)frustration, d)mental demand, and b) word per minutes for each condition for each survey prompt starting from the initial baseline prompt.

*'dear' to 'dead'."* Participants 2, 8 and 14 indicated that they sent a professional email to their employer and autocorrect changed some words to common slang terms. Participants 2 indicated that he sent his boss a curse word by accident because he used his phone to send an email. On the positive side, autocorrection can also lead to unexpected pleasant outcomes, including for Participant 4 who indicated that his friend got married because of an autocorrection changing *"have"* to *"love,"* in a situation where the recipient seems to already have been in love with his friend.

We asked also how autocorrection makes participants feel. Some expressed positive emotions such as good, happy, confident comfortable, easy, safe, satisfied, less stress, and *"makes life easier."* Yet, others mentioned negative emotions such as frustrated, irritated, aggravated, bothersome, annoying, lazy, and unsatisfied. Some were neutral and indifferent. Participant 11 and 18 mentioned that autocorrection *"weirded them out"* because autocorrect can present sensitive data, such as passwords or names, which should not been stored, or personal suggestions that they do not recall typing into their phone.

Additionally, we asked participants about the type of words that are hardest to correct after an incorrect autocorrect. Participants mentioned errors due to grammar, especially tenses, mistakes in longer than average, complex, or new words, and surnames. Many discussed mistakes due to a forgotten space, where Participant 1 talked about an unfortunate autocorrect that happened *"when I pressed b instead of the space bar."* Four participants indicated that mistakes at the beginning of a word are usually the hardest to autocorrect. Many mentioned mistakes that occur when they use multiple languages on the keyboard.

Finally, we asked participants about their design recommendations. Participant 3 said that systems designers should *"make it slightly more hidden and less distracting,"* while Participant 8 said, *"I think if we made mistakes on typing there [should be] a sound like [an] alarm, it will be useful"* and suggested that *"Highlighting [the] background of suggestion[s]"* might be helpful.

Many participants mentioned that they would prefer if there were a button on the keyboard to quickly toggle autocorrection in a single click, instead of having to go into the settings dialog. Participant 18 added, *"I think you should have a confidence score on the side of the screen so users could feel comfortable turning it off at times."* Participants 5 and 19 indicated that they want to see synonyms, one of which suggested, *"Maybe keep the drop-down option or even add it to the screen while typing with various spellings or adding an option to [show] a meaning or similar words [thesaurus option]."* Participant 10 suggested allowing deletion of standard dictionary words: *"I have no idea what a 'wyeth' is, but it's in my Android dictionary and can't be deleted."* Others suggested sentence completion using artificial intelligence. Finally, Participant 9 expressed a desire for an option for autocorrections based on their location, as people communicate differently in different geographical locations.

After they completed the task, we asked participants about their text entry behavior during our tasks. A majority, 70% indicated that they typed as fast as possible, while 30% reported that they were as careful as possible. All participants entered text using (the thumbs or fingers of) both hands.

### 5.0.4  Observations During Text Entry

We reviewed the videos from the experiment to further understand user behaviors. We saw that expressions of frustration were much more frequent with conditions where more autocorrection failures occurred; However participants were less expressive about their frustration in the conditions with 10% and 20% failures, compared to no autocorrection. Participants that experienced no autocorrection freely expressed their frustration and let us know about their feelings. We also found that our experiment was quite sensitive to user

behaviors. For instance, we identified two spikes in the reported frustration for an optimal condition participant. Going back to the videos we observed that they had said *"the word 'distraction' is really hard to type"* and in the other instance, they mentioned that typing the word *"rectangular"* was time-consuming for them. Another participant with the optimal condition said, *"I am not sure; I am confused about the autocorrection. I want to go back and fix a mistake, but it is fixed for me [pause], which is good by the way."* Yet another optimal-condition participant mentioned that they did not know if the autocorrection was on, but their frustration level was low for the whole session. Two participants experiencing the condition of autocorrection with 10% mistakes said that the *"autocorrection feature here [is] very similar to what I have in my phone."* At the beginning of the study, we also observed that the majority of our participants did not know how to turn their autocorrection off, i.e., we had to help them turn it off. This did not apply to those who used custom keyboards.

## 6 DISCUSSION

We see some evidence that perfect autocorrect is better than other autocorrection alternatives. Furthermore, autocorrect that fails 10% of the time is in some measures better than 20% failures, which in turn is also generally better than no autocorrect. Overall, as Figure 5 illustrates, lowering the percentage of autocorrect failures will reduce frustration.

We observed that using autocorrection significantly increases typing speed compared to not using it, with the optimal option being the fastest. However, we did not find a statistically significant difference between failing 10% and 20% options in terms of typing speed. When we compared the typing speed for each condition, we noticed that the participant's speed increased over time during our experiment, while without autocorrection it initially increased but then flattened out, see Figure 7.

Participants spent the least time verifying the phrases in both the optimal and autocorrection with 10% failing conditions, see Figure 4. That is explained by our finding that both conditions were not significantly different in terms of both mental demand and effort, see Figure 5.

The significantly higher number of keystrokes per character without autocorrection provides supporting evidence that the condition without autocorrection significantly decreased participants' typing speed, compared to all other options for autocorrection, see Figure 4. This also matches results from previous work (e.g., [5])

No autocorrection and autocorrection with 20% failures stood out as the most frustrating conditions. There's also a chance that the frustration stems from the participants' frustration with themselves for making errors, instead of frustration with the autocorrection itself. A participant noticed an autocorrection error and said *"I am very bad typer, I never fix my mistakes [pause], maybe it is just me."* This may be due to the fact that frustration can also lead users to believe that they are failing at a task [7]. This raises the question of how small the acceptable percentage of autocorrection failures should be. This is an interesting avenue for future, quantitative studies.

Despite occasional failures, participants felt that they had less mental demand with autocorrection regardless of its accuracy, see Figure 4. In the post-session interview, we asked them about their behaviors and perceptions around autocorrection. Most of them said in one way or the other that they accepted that autocorrection fails occasionally. As previous research has identified, lowering expectations can make emotional responses to subsequent failures less intense [44].

The physical demand significantly increased based on autocorrection accuracy, see Figure 5, since more frequent mistakes require more editing, which increases physical demand. Also unexpectedly, physical demand peaked with no autocorrection. Mental and physical demand, as well as frustration, all exhibit similar patterns, with

the optimal condition being the least demanding and no autocorrection being the most demanding condition, see Figure 7.

Participants felt that they needed to spend less effort on completing the task in the optimal condition, see Figure 5. Interestingly, and in contrast to the other conditions, effort generally decreases over time.

Participants indicated that autocorrection is overall a useful feature, when used sensibly. However, they also felt that it can sometimes change the meaning of a sentence entirely if they do not pay sufficient attention. As mentioned in the description of the experiment, when we asked for stories about positive or negative episodes that participants had experienced with autocorrection, participants said that autocorrect sometimes produces hilarious mistakes such as *"my friend was complaining about autocorrect in a text and it was changed to 'auto cucumber'."* However, some indicated that autocorrection can lead to serious mistakes and social embarrassment (see Section 4.4.3). Thus, participants said that in certain scenarios, e.g., sending professional emails or texting parents, they have to verify the text a couple of times and be more cautious. With the advance in algorithms and personalization, users are sometimes exposed to side-effects, which can save sensitive data that is then shown at inappropriate times, such as specific words that they use only in contexts unrelated to the current text message (e.g., passwords). Some participants mentioned that autocorrection *"weirded them out"* and that they were concerned about potential privacy issues.

Our participants generally indicated that the autocorrect mistakes that are hardest to correct are the ones that happen at the beginning of a sentence, likely because it takes longer to navigate to such positions in the text. There is substantial research on how to facilitate error correction, and many keyboards provide advanced techniques to tackle such issues, e.g., WiseType [1] or other work [2, 43]. However, most of these techniques have not yet been adapted in built-in keyboards on most smartphones. Many participants indicated that mistakes occur when they use multiple languages on the keyboard, which was fairly prevalent in our participant pool. There is thus a need to re-consider how multiple dictionaries should be handled as well as better language detection methods within a keyboard's implementation. Also, our participants emphasized that they would like to see keyboards with a built-in grammar checker. Grammar checkers are not yet widely available on commercial mobile keyboards at the time of our work, but recent work found that adding a grammar checker helps improve text entry speed and accuracy [1].

Participants were split about how they prefer visual feedback for autocorrects that occurred in the text. Some wished to have slightly more hidden and less distracting feedback, while others wanted highlighting and more obvious feedback for autocorrects. This indicates the importance of giving users the ability to change the visualization settings for autocorrection instances, not just the option to turn it on/off.

Participants made several interesting design recommendations. Many participants indicated that they would prefer if there were a button on the keyboard to quickly toggle autocorrection using a single step, instead of having to go into the settings dialog, see Figure 9. Some existing virtual keyboards have an option to turn off the autocorrection. However, this always requires multiple interaction steps through settings dialogues and similar mechanisms. The majority of our participants indicated that they did not know how to turn autocorrection off and on. One mentioned the idea of having a confidence score on the side of the screen. Others indicated that they want to see synonyms as drop-down options, similar to some desktop text processing systems. Another recommendation is to have an option for autocorrections based on the current location, because people communicate differently in different areas, i.e., the requirement for correctness is typically higher at work. Many participants said that they did not know how to delete words from dictionaries, which demonstrates that there are more opportunities

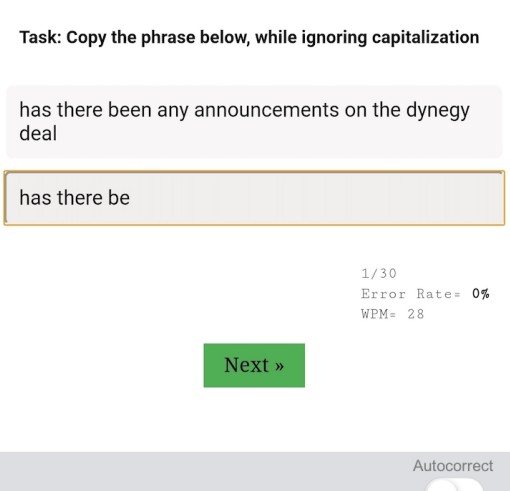

Figure 9: A design recommendation from our participants for adding a button on the keyboard to quickly toggle autocorrection.

to improve the interaction with the dictionary supporting autocorrect (for more details see section 4.4.3).

Even though we collected data from only five participants per condition, the significant differences in our results exhibit large or (at least) medium effect sizes, which we see as an indication that our results are unlikely to be spurious. Also, we point out that Kapoor et al.'s research on automatic prediction of frustration in an intelligent system relied similarly on only four participants per condition [29].

A potential limitation of our work is that our autocorrection implementation might have produced different outcomes relative to system-generated predictions, which are typically based on machine-learning-based approaches [51]. Yet, as autocorrect works differently on different platforms, we could not identify a simple way to perfectly match the behaviour that users are used to across platforms, while still giving our software access to uncorrected input and/or allowing us to implement an optimal autocorrect condition. Also, two participants stated of their own volition, i.e., without prompting or questions from our side, that they perceived our 10% failing autocorrect implementation to match closely the one on their current smartphone. One reason behind this is that many users use smartphone models that are a few years old, which means that their experience with autocorrect also lags behind the state of the art, especially on the Android platform, which many participants used. Thus, we believe that we can still state that at the time our study was performed, our implementation was ecologically valid for the study.

Additionally, we used our own implementation because we wanted to tightly control the percentage of autocorrect failures and to explore the best-case scenario with "perfect" autocorrect conditions, which is similar to the "100% accurate" autocorrect condition in [5]. This condition closely resembles an oracle. Even with the use of advanced predictive autocorrection algorithms, it would be impossible to guarantee that a given number of failures would occur, especially since we cannot predict when or how the user enters any misspelled word. After all, wrong autocorrections can be due to participants entering unrecognized words with (potentially compounding) issues,

such as spelling the word wrong, using the wrong touch locations, and/or missing a space. Interestingly, powerful autocorrect algorithms that predict corrections based on words, sentences, and user history can fail as well. Some of our participants that had the newest phones among our participants indicated that the quick adaptability of these newer methods can create issues for them, such as the system memorizing slang or curse words and then ranking them highly, in a situation where participants do not want the system to utilize such content for autocorrection.

## 7 CONCLUSION

We assessed the effect of autocorrection failures on the user's mental and physical demand, performance, and effort during typing tasks using self-report measures, a think-aloud protocol, and interviews. We showed that the higher the frequency of autocorrection failures, the more likely it is that participants become frustrated. Then we listed several design recommendations for giving users the ability to temporarily adjust the behavior of autocorrection.

In the future, we will conduct a study to explore the effect of methods that are designed to ease users' frustration when autocorrection fails. We also want to identify behavioural patterns around user frustration and potentially conduct quantitative studies that pinpoint at which failure percentage the frustration associated with autocorrect disappears. Finally, we also plan to look further into how to better support autocorrection for bilingual users and the implications of autocorrect failures that occur when using multiple languages on a keyboard.

### ACKNOWLEDGMENTS

We would like to thank the participants. The work was funded by King Saud University to whom we are also grateful.

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
