# OpenReview forum: "Auto-Cucumber: The Impact of Autocorrection Failures on Users' Frustration"
_graphicsinterface.org/Graphics_Interface/2022/Conference — GI 2022_

### Official Review · Reviewer_fHBQ · 2022-01-04
**Soundly conducted study but applicability of results could be stronger**

**Rating:** 6
**Confidence:** 3

**Review:**

This paper presents a study measuring users’ frustration level in relation to varying level of autocorrection support. While like all experimental studies, the design and execution could have been stronger (e.g., more participants), the study was overall conducted rigorously, and the results are interpreted appropriately given the study design.

A weakness of the paper is however in the motivation and framing of the applicability of the results. In this vein, there are a few areas that could be improved to help make the contribution clearer.

First, there is very shallow review of past research into autocorrect. While the paper correctly points out that most of these works have focused on speed and accuracy and not frustration, it still seems important to review that foundational work. One place where this particularly becomes problematic is in the discussion where mechanisms are proposed for better autocorrect features. Recommendations and novel interface supports have been proposed in prior work as well. And at least some research has studied the value of such supports (http://graphicsinterface.org/proceedings/gi2019/gi2019-4/). The lack of discussion of these works makes it hard to assess the value and originality of the proposals offered in this paper.

Second, the paper focuses a lot on the differences between accuracy conditions. This seems somewhat obvious as the conditions are directly manipulating workload and the only thing changing is the amount of support provided. The more interesting part of the research is the comparison between no autocorrect and autocorrect as no autocorrect is more than just 0% accuracy autocorrect (although the paper at times seems to be framing it as such). While ‘no autocorrect’ does increase workload by not providing correction support, it also increases user control and predictability and results in a different kind of error (i.e., autocorrect forces everything to be a real word which sometimes turns an incorrect but completely understandable typo into something that is completely incomprehensible – or worse misleading). As such, it’s not clear that ‘no autocorrect’ will be more frustrating than ‘autocorrect’ and it is particularly compelling that even poor autocorrect (20% failure) was less frustrating) than no autocorrect. It seems a missed opportunity to not focus more on that comparison.

Third, related to point 2 some parts of the paper were a bit unclear. For example, it’s stated that participants were free to use their own keyboards, but some keyboards (e.g., swipe) are dependent on a predictive algorithm so presumably there were limits on what keyboards could be used. I’m also a bit unclear about how user behaviour / performance intersected with the autocorrection accuracy. I would imagine that participants who typed more accurately would have experienced fewer errors (i.e., in the 20% failure condition, it’s 20% of the participant’s errors that are mis-corrected and thus the participant’s error rate dictates the absolute number of errors in need of correction, right?) I would have minimally expected to see this reported if only to establish that there were comparable ranges of ‘user performance’ across the conditions.

Finally, there is no discussion of the ‘quality’ of an error and how errors are not all equal (this is also somewhat related to point 2). To take the example presented in the discussion of ‘have’ --> ‘love’, not all errors are equally frustrating and it’s not clear how this was captured in the experiment. In the case of have->love, we might assume that the user originally typed something like ‘lave’ and that autocorrect decided to go a->o instead of l->h. This results in two costs: (1) the effort to fix the input increases (two characters to change instead of one) and (2) the potential for misunderstanding. While the case in the paper had a happy ending, this is not always true, and it seems much of the frustration with autocorrect in the wild, stems from this. Left as ‘lave’ the recipient would clearly know that there was a typo and would be left to either ask for clarification or guess at the intended meaning. With autocorrect, the recipient has less indication that there has been an error. The artificial nature of the experiment would have underplayed this aspect as participants didn’t need to fear misunderstanding or embarrassment. It might be hard or impossible to really capture this in a controlled experiment, but I think it should have at least been thoroughly discussed.

Overall, I marginally lean to acceptance on this paper. While I find it a bit hard to unpack the contribution, I think the core research is sound. I’m thus willing to leave it to the future to sort out how these results can be used. But I also think that with a bit further development of the ideas in the paper, it could be improved to make a much stronger contribution, and thus there would be value in not accepting the paper at this time to make space for that work.

---

### Official Review · Reviewer_Zzmc · 2022-01-14
**fine work, but unsure about the significance of the contribution**

**Rating:** 6
**Confidence:** 4

**Review:**

This paper presents a study about the impact of autocorrection errors on smartphone text entry, predominantly focusing on users’ frustration and workload, but covering performance as well.

I believe that the great majority of users are quite familiar with the limitations of autocorrect. It can be helpful, but it is also frustrating as it gets things wrong some of the time. Sometimes the user sees the incorrect autocorrection and has to spend the time to fix it, other times it goes unnoticed and results in what can sometimes be humorous or embarrassing outcomes.

Given that many people are familiar with this issue, I was left wondering about the strength of the contribution. It is likely novel, in that, to the best of my knowledge, others have not published a comparable study before. But I am not sure it is terribly significant, as the results are largely what I believe people would expect. As accuracy goes down, frustration and workload go up.

I had hoped that the work would have gone deeper into the autocorrection problem. For example, I wondered what are representative real-world accuracy levels perhaps for current mainstream phones vs. state of the art ML prediction (that the paper argues are not being widely used yet). Does the difference in accuracy between these two result in a significant difference in frustration and workload? Is there a tradeoff point at which autocorrection is still benefiting performance but the frustration becomes too high such that it is just not worth it? If so, does that point differ across users?

In terms of methodological approach, I appreciated that the authors took a mixed-methods approach. But I was left wondering what were the surprising findings/takeaways from the Interviews(5.0.3) and Observations (5.0.4).

The design recommendations did not feel overly strong to me. For example, the idea to surface a setting to toggle autocorrect on and off on the one hand seems reasonable and straightforward, but on the other hand, is going to take valuable real estate. It is common for users not to know where settings are buried (this is not at all unique to an autocorrect setting), and it seems that we cannot simply surface them all to top level to address the issue.

Minor: I didn’t quite understand Figure 1 in terms of why “how” was typed into the text box, given that it isn’t close to the phrase to be transcribed.

The writing was generally good, but I felt that the framing in the Introduction and in the Related could have been stronger/tighter. There was some repetition in places, for example, the design recommendations paragraph in the Discussion, largely seemed to duplicate what had already been covered in the results.

---

### Official Review · Reviewer_yBRA · 2022-01-15
**Better autocorrect is less frustrating**

**Rating:** 7
**Confidence:** 4

**Review:**

This paper presents a study of manipulating the accuracy of autocorrect to assess the impacts on performance and experience (frustration, physical and mental workload). The study seems carefully designed and controlled, having participants type a series of phrases on their own familiar keyboard, while using an interface which manipulates the accuracy of corrections. Realistic levels of autocorrect accuracy (i.e. high accuracy) were applied. The results showed generally predictable patterns: without autocorrect mental and physical workload was higher, as was frustration. Performance was degraded (wpm, KSPC, etc.). The more accurate the autocorrect, the lower the workload, frustration, and the higher the performance (though statistically significant differences were not found between all steps).

The paper is clearly written and the results are presented in a standard way. Some of the figures are not designed well. For example, the labels on Figure 6 are too small to read without enlarging. The learning effects evident in Figure 6(e) are not really discussed, but they seem to signal to me that perhaps the first few trials should have been discarded as performance had not stabilized. Another drawback is that the autocorrect used in the study was deficient compared to standard keyboards in Android and iOS, which provide a preview of autocorrect / autocomplete in real-time while typing. Given the performance degradation when autocorrect is turned off, I'm dubious about the suggestion of a switch to deactivate it right on the keyboard. I think the harm of forgetting to re-enable it would be worse than the problems it causes, but perhaps that is an avenue for future investigation.

It wasn't clear to me why all the input text was lowercased. I guess this evened out the trials, but in realistic typing, case differences can be informative to autocorrect systems (e.g. in ignoring proper nouns like 'dynegy' presented in Figure 7). Another concern I have is the use of a think-aloud protocol simultaneous with a timed task. Couldn't it be that participants thinking aloud would have lower typing performance and/or higher mental load? Or was think aloud only applied between trials? Finally, there is a missed opportunity here to address the autocorrect error that I expect many people find frustrating - erroneously correcting non-words (slang, proper names) to real words. This would likely offer an advantage to the no correct condition. I guess that's one argument for the toggle, but it would have been interesting in the study to have introduced some of this sort of autocorrect failure.

Overall, this is an acceptable paper. The results are predictable, but of course, that is part of the practice of science and we shouldn't hold that against the work. It was interesting to see that some participants were distracted by the 'too perfect' optimal autocorrect, but in practice, I expect they could get used to that, and performance would remain high overall.

Minor:
Figure 1 - why is 'how' in the box? This doesn't seem close enough to the target.
Figure 4 - define what the levels of significance are for the asterisks
Section 6: "section 4.4.3" -> "Section 4.4.3"
Figure 4 and Figure 5: be consistent in the naming of conditions, e.g. "10% Failing" vs "Fail 10%"

---

### Decision · Program_Chairs · 2022-01-18

Accept